# Recent Advances in Experimental Functional Characterization of GWAS Candidate Genes in Osteoporosis

**DOI:** 10.3390/ijms26157237

**Published:** 2025-07-26

**Authors:** Petra Malavašič, Jasna Lojk, Marija Nika Lovšin, Janja Marc

**Affiliations:** 1Faculty of Pharmacy, University of Ljubljana, Aškerčeva cesta 7, 1000 Ljubljana, Slovenia; petra.malavasic@sb-nm.si (P.M.); marija.nika.lovsin@ffa.uni-lj.si (M.N.L.); 2Department for Laboratory Diagnostics, General Hospital Novo mesto, Šmihelska cesta 1, 8000 Novo Mesto, Slovenia; 3Clinical Institute of Clinical Chemistry and Biochemistry, University Clinical Center Ljubljana, Njegoševa cesta 4, 1000 Ljubljana, Slovenia

**Keywords:** in silico analyses, omics, MSC, human bone tissue gene expression, gene knockdown

## Abstract

Osteoporosis is a multifactorial, polygenic disease characterized by reduced bone mineral density (BMD) and increased fracture risk. Genome-wide association studies (GWASs) have identified numerous loci associated with BMD and/or bone fractures, but functional characterization of these target genes is essential to understand the biological mechanisms underlying osteoporosis. This review focuses on current methodologies and key examples of successful functional studies aimed at evaluating gene function in osteoporosis research. Functional evaluation typically follows a multi-step approach. In silico analyses using omics datasets expression quantitative trait loci (eQTLs), protein quantitative trait loci (pQTLs), and DNA methylation quantitative trait loci (mQTLs) help prioritize candidate genes and predict relevant biological pathways. In vitro models, including immortalized bone-derived cell lines and primary mesenchymal stem cells (MSCs), are used to explore gene function in osteogenesis. Advanced three-dimensional culture systems provide additional physiological relevance for studying bone-related cellular processes. In situ analyses of patient-derived bone and muscle tissues offer validation in a disease-relevant context, while in vivo studies using mouse and zebrafish models enable comprehensive assessment of gene function in skeletal development and maintenance. Integration of these complementary methodologies helps translate GWAS findings into biological insights and supports the identification of novel therapeutic targets for osteoporosis.

## 1. Introduction

In recent decades, different methods for high-throughput genomic screening have been developed and implemented, such as genome-wide association study (GWAS), which is an approach based on mapping the variation in genomic sequence to find associations between genetic variants and selected phenotypic traits. This approach is especially useful for diseases or traits with no clearly defined genetic components and for multifactorial diseases, which result from a combination of environmental factors and changes in regulation of several genes simultaneously. One polygenic disease is osteoporosis, an age-related chronic skeletal disease, characterized by decreased bone mineral density (BMD) and defects in bone microarchitecture, resulting in decreased bone strength and a higher risk of bone fractures. Osteoporotic fractures dramatically reduce the quality of life and lead to the accelerated progression of other geriatric diseases or even death, thus representing a growing health and socio-economic burden [1]. As osteoporosis affects more than one third of postmenopausal women and, to a lesser extent, also older men [2], a better understanding of the mechanisms and biology of osteoporosis is crucial for improving current preventive, diagnostic, and treatment approaches.

The first GWAS using BMD and osteoporosis as the study trait was published by Kiel and co-workers in 2007 [3]. In the following years, several consortia formed worldwide, which led to a significant increase in study population size, but the largest increase in the number of recognized loci was achieved with collaborations and large-scale GWAS meta-analyses. To date, more than 1000 associated loci have been identified, and they are estimated to explain around 20% of the variance in BMD [4]. However, identification of candidate genes is only the first of the many steps required to confirm the involvement of the identified gene in the studied diseases and explore its function in disease biology (Figure 1). The actual causal gene may be different from the candidate gene, and this can only be determined with functional analysis. Using suitable cellular, animal, and molecular methods, it is possible to evaluate the mechanisms of each candidate gene in bone pathology to reveal how gene combinations contribute to the development of osteoporosis [5]. This could further lead to confirmation or discovery of novel proteins involved in the disease pathogenesis; a better understanding of the genetic architecture of the disease, susceptibility, and predisposition; identification of new biomarkers; therapy optimization; and identification of possible new drug targets for osteoporosis [4,6]. Unfortunately, most SNPs associated with osteoporosis have been found in non-coding intergenic and intronic regions or were associated with genes with so far unknown involvement in osteoporosis, and only a small fraction of detected loci have been successfully characterized so far [4,7,8].

The goal of this review is to examine current methodologies and key examples of the successful functional characterization of GWAS-identified candidate genes in osteoporosis research. With this work, we aim to bridge the methodological gap between genetic variation discovery and biological validation in osteoporosis research by providing clear guidance and a goal-oriented framework to help researchers select appropriate tools for functional gene validation tailored to the context of bone biology. By leveraging the methodological overview and practical examples presented in this review, scientists will be better equipped to choose and combine experimental strategies that accelerate the discovery of how specific genes and their protein products influence bone cell physiology, contribute to skeletal tissue homeostasis, and ultimately play a role in the development of complex disorders such as osteoporosis [9,10]. With this we hope to facilitate and encourage the urgently needed experimental studies on confirmation and functional characterization of candidate genes that move beyond statistical associations and prediction models. Importantly, this review specifically focuses on the evaluation of gene function itself—independent of variant-specific regulatory effects—and does not address functional studies of non-coding regulatory variants, transcript isoforms, or variant-driven changes in gene expression. These topics are comprehensively covered in other reviews [11,12,13].

## 2. Osteoporosis as a Multifactorial Disease

Primary osteoporosis develops as an age-related imbalance in bone formation (mediated by osteoblasts) and bone resorption (mediated by osteoclasts), which is a constantly occurring turnover process required for bone growth, repair, and adaptation to increased or decreased bone loading. These processes are mediated through a complex network of signaling pathways and factors, such as Wnt/Hippo, Notch, JAK/STAT, NF-κB, and TGF-β signaling pathways, growth hormone/insulin-like growth factor-1 (GH/IGF-1), and steroid hormones [14]. This cell signaling is highly regulated through an intricate network of positive and negative feedback loops, both at the protein and transcriptional levels, which relies on the correct function of all these components, all of which are subject to genetic variability. BMD is thus highly polygenic in nature, which means that the occurrence and development of osteoporosis are determined by many gene variants, each with a small contribution to the final outcome, with heritability estimates of 0.6–0.85 [15,16]. Moreover, BMD might also be influenced by fetal or maternal genetic and epigenetic factors, events, and environmental influences during embryonic development and childhood, when the skeletal size and density increase at the highest rate [17]. Therefore, BMD is the sum of complex interactions between environmental factors (such as diet, physical activity, medication), co-morbidities, and genetic background. Risk factors for osteoporosis include older age, female sex, menopause, family history, low BMI, low serum vitamin D and calcium levels, sedentary lifestyle, smoking, alcohol consumption, physical inactivity, and the use of various medications such as corticosteroids, anticonvulsants, proton pump inhibitors, and others [18].

Unfortunately, due to a lack of specific symptoms, osteoporosis often goes undiagnosed until the first osteoporotic fracture occurs. This highlights the critical need for functional studies of genes identified through GWAS to elucidate the physiological and pathological mechanisms underlying bone diseases. Such studies are essential for the discovery of novel biomarkers for osteoporosis and for advancing our understanding of pathogenesis, ultimately enabling more effective prevention, diagnostic, and treatment strategies.

## 3. Osteoporosis Genome-Wide Association Studies

In osteoporosis GWAS, the observed trait is usually site-specific areal BMD (hip, heel, spine, femoral neck, forearm, total body BMD), which is obtained from dual-energy X-ray absorptiometry (DXA), but other imaging methods have also been used, such as quantitative computed tomography (QCT) or quantitative ultrasound (QUS) [19,20,21,22]. After the trait is defined, a cohort of osteoporotic patients and a well-matched group of healthy individuals are selected. Genotyping can be performed using commercial single-nucleotide polymorphism (SNP) arrays (combined with statistical imputation) or whole-genome sequencing (WGS) for both groups, followed by statistical and association tests to identify variants in specific genomic regions (loci) with a statistically significant difference (dependent on the sample size, typically *p* ≤ 5 × 10^−8^) in frequency compared to healthy individuals. However, these genotyped regions are usually not the direct causal variants, but are in linkage disequilibrium with them, so additional sequencing is performed in that particular locus to identify the exact genetic change involved in the observed trait [6,23,24]. Most association hits actually point to non-coding regions of the genome, and due to the complex chromosomal organization and distal regulatory regions, functional studies suggest that only one third of causal variants correspond to the nearest gene to the detected GWAS hit [25,26]. This makes gene annotation and functional characterization of genetic variants both challenging but necessary in order to confirm the role of observed SNPs.

The first GWAS of osteoporosis was published by Kiel and co-workers in 2007 [9]. Due to the small sample size, the identified SNPs had low *p*-values (*p* > 7 × 10^−5^) and did not reach the currently agreed-upon threshold of *p* ≤ 5 × 10^−8^. Nevertheless, the study recognized several SNPs in candidate genes that had already been studied for their role in osteoporosis and set a precedent. The following GWA studies on BMD increased the sample size, but the largest increase in the number of recognized loci was achieved with collaborations and large-scale GWAS meta-analyses. The first meta-analysis was performed by the GEFOS consortium and, with 19,195 samples (discovery), identified 13 novel loci [27]. Three years later, the meta-analysis was expanded to 32,961 samples (discovery) and, besides replicating the majority of already known loci, identified an additional 32 SNPs associated with BMD in femoral neck and lumbar spine [9]. The largest series of GWASs for BMD was performed on heel ultrasound BMD measurements on participants in the UK Biobank Study only recently. In 2017, Kemp and co-workers associated heel BMD with 203 loci with a genome-wide significance threshold of *p* ≤ 6.6 × 10^−9^ in 142,487 samples, of which 153 loci had not previously been implicated in BMD in GWASs [22]. Shortly after, the sample size was increased to 426,924, yielding 518 loci, out of which 301 loci were novel [19]. In a series of alternative approaches to data analysis of the UK Biobank Study data that followed the initial studies, several additional novel loci were identified [20,21,28,29,30]. Despite the large sample size achieved, especially with meta-analyses, it is estimated that the power limitation still conceals additional rare associations or associations with smaller effect sizes, which could explain the still-missing part of trait heritability as estimated by twin and family studies [31]. To date, more than 1000 associated loci have been identified, which are estimated to explain around 20% of the variance in BMD [4].

Detected genomic variants must be functionally annotated, and in the case of non-coding variants, the most probable target gene must be identified. To achieve this, numerous computational prediction tools have been developed for quality control, genotype imputation, association testing, functional gene annotation, mapping, and enrichment analyses [11]. SNPs are fine-mapped and annotated using databases such as ENCODE and ATAC-seq to predict disease-causing variants. In cases where genetic signals are significantly (*p* < 10^−8^) associated with specific bone phenotypes, tools like DEPICT are used to predict gene involvement in bone-cell-specific pathways [32]. The final result of a GWAS is thus a list of candidate genes that must be evaluated and confirmed, which can be achieved through the process of functional characterization.

## 4. Post-GWAS In Silico Studies

Once candidate variants are identified through GWAS, in silico analyses play a crucial role in predicting their functional relevance and guiding experimental validation. These computational approaches allow researchers to explore how genetic variants may influence gene regulation, protein structure and function, and broader biological pathways. One of the first steps in post-GWAS analysis is statistical fine-mapping, which helps narrow down genomic loci to the most likely causal variants. Tools such as SuSiE, FINEMAP, CAVIAR, and PAINTOR estimate the probability of individual variants being causal and define credible sets that are highly likely to contain the true causal variant [33,34]. Functional annotation and regulatory mapping provide further insight, particularly for non-coding variants. Expression quantitative trait locus (eQTL) analyses using resources like GTEx, eQTLGen, and sc-eQTL can reveal whether genetic variants affect gene expression levels [35]. Colocalization tools such as COLOC, eCAVIAR, and SMR are employed to determine whether GWAS signals and regulatory effects share a common causal variant. In addition, chromatin conformation data obtained from Hi-C, Capture-C, or ChIA-PET experiments, available through platforms like the 3D Genome Browser and ENCODE, help establish physical interactions between distant regulatory elements and their target genes. Epigenomic annotation tools, including HaploReg, RegulomeDB, ChromHMM, and DeepSEA, provide valuable information about the chromatin state, transcription factor binding sites, and the predicted regulatory impact of individual variants [26,36,37]. For coding variants, in silico tools such as PolyPhen-2 [38], SIFT [39], MutPred [40], and advanced protein structure prediction models like AlphaFold [41] can be used to assess the potential impact of amino acid substitutions on protein structure, stability, and function. Genetic variants may influence epigenetic mechanisms such as DNA methylation. These effects can be explored using methylation quantitative trait locus (mQTL) databases like mQTLdb and integrative resources such as EpiMap, which provide data on the interplay between genetic variation and epigenetic modifications [42,43]. To understand the broader biological implications of GWAS findings, pathway and network analyses are conducted using tools like MAGMA [44], DEPICT [32], and GSEA [45], which identify enriched biological pathways and gene sets associated with the identified variants. Protein–protein interaction networks, accessible via platforms like STRING [46] and GeneMANIA [47], further support the functional interpretation of candidate genes by highlighting their interactions within biological systems. Finally, the integration of GWAS results with multi-omics datasets, including transcriptomics, proteomics, and methylomics, enables a more comprehensive understanding of the downstream effects of genetic variation. Polygenic risk scores (PRSs), calculated using tools such as PRSice-2 and LDpred, help quantify the cumulative effect of multiple variants on disease risk [48,49,50].

## 5. Experimental Functional Characterization of GWAS Hits

Once the causal variants have been fine-mapped and functionally annotated, different laboratory experimental approaches can be used to test and confirm the in silico-generated predictions and determine the actual effects of these variants on bone biology. The choice of approaches depends on the predicted function of the gene, based either on known roles of the same gene/protein in other tissues or on the function of proteins of the same protein family (e.g., structural protein, signaling factor, receptors or ligands, proteins with regulatory/modulatory function, etc.). Depending on the function of the gene/protein, different cellular processes can be expected to be affected, such as cell metabolism, differentiation, attachment, proliferation, and function. Any of these can result in changes in bone cell behavior and homeostasis, which can further lead to changes in bone tissue function and quality. This predicted function of the gene of interest will thus determine the best approach to its functional characterization, starting with the adequate choice of a cellular or animal model, best approach for gene/protein expression alteration, and of course the most informative methods for assessment and quantification of the resulting changes in different aspects of bone cell biology (Figure 2). For example, functionally characterizing a gene predicted to play a role in cell differentiation will require a choice of a cellular model capable of differentiation, while a gene with a role in cell function might require a cell model that can mineralize. Similarly, while the function of certain genes would be best explored with stable knock-out cell lines, other genes involved in essential cell processes or proliferation can only be studied by means of transient or inducible changes in their expression. The following sections will address each of these approaches separately in more detail.

### 5.1. Selection of In Vitro Cell Model

The choice of an appropriate in vitro cell model is essential for accurately characterizing the functional effects of GWAS-identified variants. Primary cells, such as human mesenchymal stem cells (MSCs) and primary osteoblasts, closely mimic physiological conditions and maintain natural differentiation potential, making them highly relevant for studying osteogenesis. However, their limited lifespan, donor variability, and lower suitability for genetic manipulation present practical challenges. Immortalized cell lines offer easier handling and unlimited proliferation and are more amenable to genetic editing techniques like CRISPR/Cas9. Despite these advantages, they often display altered cellular phenotypes, limited differentiation capacity, and reduced physiological relevance due to their transformed or tumor-derived nature. Model selection should be based on the specific research question, balancing physiological relevance with experimental feasibility. A combined approach is often recommended—using immortalized cell lines for initial mechanistic studies and confirming findings in primary cells [51,52,53].

The most robust models for studying bone-related genes are immortalized human bone-derived cell lines, such as HOS, Saos-2, MG-63, U2OS, OHS-4, TE-85, TPXM, and CAL-7 (Table 1). They provide consistent, reproducible results and a long-lasting supply of cells for experiments. However, it is important to recognize that these cell lines may not fully recapitulate the complexity of normal bone physiology or reflect the entire spectrum of bone diseases. These limitations should be considered when using immortalized cell lines, and findings should be complemented with other experimental models such as primary cells and animal models. Immortalized cells represent the first choice for rapid functional characterization of GWAS hits. Using immortalized cell models, several genes have recently been implicated in bone metabolism (e.g., USF3 [54], ANAPC1 [55], CCDC170 [56]).

The choice of cell line should reflect the biological process under investigation (Table 1). For example, for studies focused on mineralization and late-stage osteogenesis, models such as Saos-2 and MC3T3-E1 are most appropriate. Saos-2 cells, derived from human osteosarcoma, exhibit osteoblastic properties and are commonly used in bone-related research due to their ability to deposit a mineralization-competent extracellular matrix. MC3T3-E1 cells, a clonal non-transformed cell line established from newborn mouse calvaria, are widely used for osteogenesis evaluation, demonstrating high differentiation potential and mineralization capacity. Experiments on early differentiation events, proliferation, and remodeling can be efficiently conducted using MG-63 or HOS cells [57]. MG-63 cells, human osteosarcoma-derived, are commonly used to investigate proliferation and early differentiation due to their high proliferative capacity and responsiveness to osteogenic stimuli. HOS cells, also derived from human osteosarcoma, are employed in studies focusing on cell proliferation and gene expression, although their mineralization capacity is limited [58].

If possible, the use of primary cell cultures is a better option. Primary cells are not immortalized, and their behavior more closely resembles the properties of cells in vivo. Primary cells used for studies of osteoporosis are either obtained from human bone (terminally differentiated osteoblasts, osteocytes, or osteoclasts and mesenchymal stem cells) or human blood (human monocytes, which are precursor cells for osteoclasts). Tissue-derived MSCs are multipotent progenitor cells that play a significant role in bone regeneration and fracture repair. In vivo, they can differentiate into bone (osteoblasts), fat (adipocytes), muscle (myocytes), or cartilage (chondrocytes) cells, making them an excellent model for investigating the function of musculoskeletal-related genes. Human MSCs are the most relevant cell model for studying human osteoblast differentiation and bone metabolism. Many studies have used human MSCs for the functional characterization of GWAS hits [59,60,61].

MSCs can be bought from all major cell-culture banks or can be isolated directly from human tissue. Commercially available MSCs, provided by biobanks and companies such as Lonza, ATCC, and PromoCell, offer the advantage of standardized quality control, validated marker expression, and known differentiation potential. These cells are typically easier to handle, ready for experiments, and ideal for laboratories without access to human tissue samples or established cell isolation infrastructure. However, the main drawbacks are higher costs and potential limitations related to donor variability and passage number, which can affect differentiation capacity. Alternatively, researchers may choose to isolate MSCs directly from human tissue such as bone. However, as human bone tissue is extremely difficult to obtain, other tissue sources are usually used to obtain MSCs, most commonly bone marrow aspirates or adipose tissue. Isolation from bone marrow aspirates involves density gradient centrifugation (e.g., using Ficoll-Paque) to separate mononuclear cells [62], followed by plating the cells under standard culture conditions to separate adherent MSCs from other cells and debris. Culturing must be accompanied by strict monitoring of cell morphology and marker expression to confirm MSC identity according to the International Society for Cellular Therapy (ISCT) criteria: positive expression of CD105, CD73, and CD90 and negative expression of hematopoietic and endothelial markers including CD45, CD34, CD11b, CD14, CD79a, CD19, and HLA-DR. Additionally, the cells must demonstrate tri-lineage differentiation potential toward osteogenic, chondrogenic, and adipogenic lineages [63,64]. MSCs derived from different tissues (e.g., bone marrow, adipose tissue, dental pulp) exhibit varying capacities for osteogenic differentiation. This variability poses a challenge in standardizing differentiation protocols and achieving consistent results across different MSC sources [65].

Another frequently overlooked aspect in bone biology research is the confirmed gender dimorphism, which results in considerably higher incidence of osteoporosis in women compared to men [66,67]. While many genes affecting BMD will behave similarly in both genders, genes directly or indirectly related to sex steroid hormones or other gender specific proteins and their many downstream targets might not. Moreover, based on the chromosomal differences between the two genders, not all genes will be present in an equal number of copies. An example is a male-specific gene SRY (sex-determining region Y), which acts as a repressor of receptor activator of nuclear factor kappa-B ligand (RANKL), a protein which stimulates osteoclast differentiation and activity, resulting in increased bone resorption [68]. Similarly, GWAS loci with potentially sex-specific influences on BMD, such as the SLC25A13 gene, have been identified [69]. The gender of the donor of the MSCs (as well as stable cell lines and animal models) should thus also be considered when selecting the appropriate cell or animal model for a certain gene. In contrast to osteoblasts and other MSC-derived cell types, osteoclasts derive from monocytes following stimulation with receptor activator of nuclear factor kappa-B ligand (RANKL). Osteoclasts are responsible for bone resorption and play a crucial role in bone remodeling, contributing to osteoporosis. Monocytes, which make up 2–10% of peripheral blood mononuclear cells (PBMCs), are a readily accessible source for generating osteoclasts in vitro. Monocytes can be isolated using advanced methods such as flow cytometry and cell sorting (FACS), which require specialized and expensive equipment [70]. More commonly, monocytes are isolated using density gradient centrifugation, for example using Ficoll-Paque, which separates PBMCs from whole blood [71,72], followed by positive selection using magnetic-activated cell sorting (MACS) with CD14-conjugated microbeads [73], or by plastic adherence methods that exploit the natural adherence properties of monocytes [74]. After isolation, monocytes are differentiated into osteoclasts by culturing them with macrophage colony-stimulating factor (M-CSF) and RANKL. During this process, monocytes undergo characteristic morphological changes, including the formation of large multinucleated cells and actin ring structures, which are typical features of mature osteoclasts. Osteoclast differentiation can be confirmed using tartrate-resistant acid phosphatase (TRAP) staining and by assessing the expression of specific receptors such as the calcitonin receptor, RANK, and the vitronectin receptor [75,76].

In GWAS studies, the *MEF2* locus was associated with osteoporotic fractures [77]. MEF2C is a transcription factor associated with the Wnt signaling pathway [78,79]. A study by Kramer et al. used osteocytes to identify the role of MEF2C in osteoclast differentiation. Knockdown of MEF2C in miceosteocytes Ps resulted in reduced osteoclast differentiation. The study demonstrated that MEF2C is a positive regulator of human osteoclastogenesis through activation of the c-Fos and NFATc1 transcription factors.

However, even primary cells do not completely replicate the in vivo organ environment [53,80]. Three-dimensional cell cultures are a step closer to the tissue complexity in vivo [81]. Three-dimensional models can be cultivated as cell spheroids [82,83] or with the use of hydrogel scaffolds [84,85]. Many differences are observed between the 2D and 3D systems, including changes in cellular architecture and extracellular matrix organization [86]. The level of osteogenesis in three-dimensional spheroid systems is enhanced compared to 2D cultures [87]. Furthermore, co-cultures of osteoblasts and osteoclasts, with or without scaffolds, can be used for functional characterization studies and cell-cell communication [88]. Advanced 3D culture systems often incorporate co-cultures of osteoblasts and osteoclasts to more accurately model bone remodeling processes. Co-cultures can be performed either in direct contact, where the two cell types are cultured together in the same space, or using indirect methods such as inserts with permeable membranes, which allow paracrine communication while maintaining physical separation [89]. One challenge in co-culture experiments is distinguishing between the two cell populations during analysis. For certain types of read-out methods, this can be addressed using cell-type-specific markers and antibody labeling (immunohistochemistry) or other staining methods. For example, osteoblasts are typically identified by the expression of ALPL, RUNX2, and osteocalcin, and their mineralization capacity can be assessed using Alizarin Red staining [83,90]. Additionally, immunofluorescence staining and flow cytometry using cell-type-specific surface markers can be employed to separate different cell types and analyze each one separately [91]. The comparison of different models for functional evaluation is shown in Table 2. 

### 5.2. Gain- and Loss-of-Function Approaches in Cell Models

Gain- and loss-of-function studies are fundamental for understanding the roles of specific genes in bone metabolism and the main mechanism used to functionally validate the candidate genes identified through GWAS. These approaches involve either reducing or eliminating gene expression (loss of function) or increasing gene expression (gain of function) in primary cell models such as MSCs and stable cell lines, as well as in animal models. While these techniques are powerful, their successful application requires careful consideration of experimental design, transfection strategies, and the limitations of the chosen cellular or animal model [92,93]. Gene knockdown (partial reduction in gene expression) is typically achieved using small interfering RNAs (siRNAs) or short hairpin RNAs (shRNAs), which suppress gene expression by promoting mRNA degradation. siRNA is used for transient knockdown [94], which is suitable for short-term studies, while shRNA (lasting for weeks or even months) allows more stable silencing when delivered via viral vectors [95]. However, these approaches have limited efficacy in long-term experiments such as osteogenic differentiation or mineralization studies, as the effects diminish over time due to a reduction in the quantity of siRNA either through degradation or due to cell proliferation. Additionally, off-target effects are a common concern, requiring appropriate experimental controls (e.g., non-targeting scrambled siRNA) and validation of silencing effects [96]. A significant problem of such transient silencing is also siRNA delivery into the cells, which is rarely 100% efficient, resulting in a heterogeneous population of cells that can also contain cells that did not receive siRNA at all. This is especially problematic when studying genes/proteins with subtle effects, which can be further diluted by the low efficacy of siRNA delivery. Transient silencing of genes is thus mainly advised for studying genes that cannot be permanently knocked out, as it would result in cell death or loss of proliferation. For the rest of the genes/proteins, stable knock-out (KO) cell lines are a better option, although they are more difficult to obtain.

For permanent gene disruption, CRISPR/Cas9 technology is widely used as it is relatively easy to design and perform (Table 3). This system introduces double-strand breaks at specific genomic loci, leading to small insertions or deletions that disrupt gene function. The easiest way to perform CRISPR7Cas9 editing is to use a commercial plasmid (e.g., pSpCas9 [BB]-2A-Puro [PX459]), which encodes the Cas9 enzyme, cloning sites for guide sgRNA, and antibiotic resistance cassettes for bacterial and eukaryotic selection [97]. A gsRNA designed to target the gene of interest can be cloned into the plasmid using commercially available restriction enzymes. Although this method is considered precise, off-target editing and the potential for introducing unintended mutations must be carefully evaluated [98]. When regulatory elements are studied or when permanent genomic changes are undesirable, CRISPR-based gene modulation systems such as CRISPR activation (CRISPRa) and CRISPR interference (CRISPRi) are effective alternatives. These approaches use a catalytically inactive Cas9 (dCas9) fused to transcriptional activators or repressors to modulate gene expression without altering the DNA sequence. This allows for reversible and tunable control of gene activity, ideal for studying dose-dependent effects of gene expression. However, these methods require high transfection efficiency and are generally limited to short- to mid-term studies unless the plasmids are stably integrated into the genome [99].

Efficient delivery of genetic material is critical for the success of these techniques, and several techniques are available based on different principles. Chemical transfection methods, such as lipid-based reagents, are easy to use and do not require specialized equipment, but often result in low efficiency, particularly in hard-to-transfect cells like MSCs and monocytes. The method is based on complex formation between the genetic material and the reagent, which binds to the cell membrane and facilitates its entry into the cell. Electroporation and nucleofection, on the other hand, use electric pulses to introduce the genetic material into the cell by creating temporary pores in the cell membrane. Therefore, electro-transfection can cause significant cellular stress and decreased viability. Viral transduction, especially using lentiviral or adenoviral vectors, enables high efficiency and stable gene expression but introduces concerns regarding biosafety and potential insertional mutagenesis [100]. It also requires a high-biosafety-level laboratory, which might not be accessible.

A critical consideration in gene modulation experiments is also whether to perform transient or stable transfection. While transient transfections are easier, quicker, and less expensive, they result in a heterogeneous cell population and are only suitable for short-term studies, as discussed previously. For longer experiments involving cell differentiation or mineralization, which require growth of the same cell population for up to a month, stable transfection is preferred, although it takes more time to prepare, evaluate, and expand the stable KO cell lines. To obtain a stably modified cell line, a plasmid (either for CHRISP/Cas9 or for overexpression) is transfected into cells that are then exposed to a selection antibiotic. Eventually, only the cells that have successfully integrated the plasmid into their genome will be able to survive and proliferate in the presence of the antibiotic. Colonies grown from single cells should then be isolated and expanded to obtain clonal cell lines, ready for verification of the obtained genetic change. Unfortunately, generating stable clones in primary MSCs is not feasible due to their limited replicative capacity and inability to grow from single cells. Immortalized cell lines are more suitable for generating stable cell populations, but their behavior may not fully reflect that of primary cells [101]. Functional studies must also consider the issue of heterozygosity, particularly when working with CRISPR-based genome editing. Incomplete knock-outs or heterozygous mutations can result in residual gene function, complicating data interpretation. As more than one clone is usually obtained, different clones should be used in experiments as biological replicates.

### 5.3. Methods and Approaches for Evaluation of Bone-Specific Outcomes

Evaluation of gene and protein function in bone-related research relies on a wide array of experimental approaches. General molecular and cellular methods such as quantitative PCR (qPCR), Western blotting (WB), metabolic and proliferation assays, immunocytochemistry (ICC), and flow cytometry are routinely employed to assess gene expression, protein abundance, subcellular localization, metabolic activity, and cellular phenotypes. These techniques are applicable across diverse biological systems and are selected based on the anticipated role of the gene or protein under investigation (e.g., transcriptional regulation, signaling, or structural function). However, to specifically assess bone-related functions, more specialized assays are required. These include in vitro models of osteoblast and osteoclast differentiation, matrix mineralization, and bone resorption, which allow functional evaluation of genes implicated in processes specific to skeletal development, remodeling, or disease pathogenesis. In this section, we will focus on experimental methods tailored to bone biology.

#### 5.3.1. Cell Differentiation Evaluation

Cell models capable of cell differentiation provide a more physiologically relevant model than undifferentiated cells, on one hand allowing the study of the dynamic processes involved in differentiation such as matrix mineralization and lineage-specific gene regulation, but such models are also a source of terminally differentiated cells, with different properties and behavior compared to undifferentiated cells. Osteogenic differentiation of MSCs progresses through distinct stages, from mesenchymal precursors to pre-osteoblasts, osteoblasts, and mature bone cells. This process is regulated by sequential gene expression: Sox9 marks early precursors, followed by *Runx2*, Osterix (*OSX*), alkaline phosphatase (*ALP*), and collagen type I (*Col1a1*). Mature osteoblasts produce bone matrix proteins such as osteocalcin (*OCN*), osteopontin (*OPN*), and bone sialoprotein (*BSP*), with *OCN* serving as a key marker of mature osteoblasts. Differentiation involves three phases: proliferation (stimulated by IGF, TGF, and FGF), commitment to the osteoblast lineage, and bone matrix mineralization. Key transcription factors include *Runx2*, which drives osteoblast differentiation; *OSX*, essential for osteoblast maturation; *DLX5*, promoting osteogenesis through signaling pathways like Notch; and *ATF4*, which supports osteoblast activity and inhibits osteoclast formation [102,103]. By culturing MSCs in vitro under specific biochemical stimuli, the process of osteogenesis and the factors regulating bone formation can be studied. Several markers are commonly used to assess osteoblast differentiation [104,105,106,107,108] (Table 4). Two newly identified genes—ING3 (at the WNT16–CPED1 locus) and EPDR1 (at the STARD3NL locus)—were functionally evaluated in human mesenchymal progenitor cell-derived osteoblasts by Chesi et al. and featured in *Nature Communications* [59]. Osteoblast differentiation was impaired after siRNA knockdown in primary human MSCs, whereas adipogenic differentiation increased. These findings highlight relevant genes involved in MSC fate determination [59].

For the study of candidate genes involved in the processes of bone resorption, osteoclast cell models should be used. Osteoclasts are multinucleated cells responsible for bone resorption, derived from monocyte/macrophage lineage precursors under the influence of specific cytokines. In vitro differentiation of osteoclasts typically involves culturing peripheral blood mononuclear cells (PBMCs) or monocytic cell lines (e.g., THP-1, RAW264.7) in the presence of macrophage colony-stimulating factor (M-CSF) and RANKL. Successful differentiation is characterized by the formation of large, multinucleated cells expressing specific markers such as tartrate-resistant acid phosphatase (TRAP), cathepsin K (CTSK), calcitonin receptor (CALCR), and nuclear factor of activated T cells 1 (NFATc1), which acts as a master transcription factor for osteoclastogenesis [66,67,109]. TRAP staining remains the most commonly used method to visualize and quantify osteoclasts, while gene expression analysis and immunocytochemistry are used to validate osteoclast-specific molecular signatures [110]. Despite well-established protocols, osteoclast differentiation experiments also face challenges. Human monocyte-derived osteoclasts are sensitive to culture conditions and prone to detachment, particularly as they form large multinucleated cells. Maintaining cell viability and ensuring complete fusion requires precise control of cytokine concentrations and cell density. Despite established protocols, differentiation experiments often face practical challenges. One common issue is the prolonged culture time required for full differentiation, leading to cell detachment and loss of samples, especially in later stages when the extracellular matrix becomes heavily mineralized. This not only affects cell viability but also complicates RNA isolation, as mineralized matrices interfere with lysis and reduce RNA yield. Another challenge is the declining differentiation potential of MSCs with increasing passage number. To maintain reproducibility, it is advisable to use low-passage cells and optimize seeding densities. In osteoclast cultures, incomplete differentiation and loss of fragile, multinucleated cells during handling are common issues. An important advantage of differentiation models in bone research is the ability to directly monitor matrix mineralization, which provides a functional readout relevant for studying bone formation and diseases such as osteoporosis. This allows researchers not only to assess molecular markers but also to evaluate the physiological outcome of gene modulation through quantifiable mineralization assays.

#### 5.3.2. Functional Assessment—Matrix Mineralization and Resorption

The process of cell mineralization, performed by osteoblasts, and bone resorption, performed by osteoclasts, are the two most important processes that define bone quality and mineral density and as such play the central role in most bone-related diseases, including osteoporosis. In a healthy bone environment, these two processes work in equilibrium, adapting the quality and strength of the bone to the requirements of the external environment, but they also play an important part in bone healing and growth. As these processes are limited to bone tissue, there is a high probability that the GWAS-detected candidate genes related to BMD will be involved in the process of bone turnover and regeneration in one way or another.

The mineralization process involves two primary phases: an initial rapid phase, during which the majority of mineral deposition occurs within a short timeframe, and a slower maturation phase, which can extend over weeks or months as mineral content gradually increases. At the cellular level, mineralization is initiated by osteoblasts through the production of type I collagen and non-collagenous proteins such as osteocalcin, osteopontin, and bone sialoprotein, which play essential roles in nucleating mineral deposition. The process proceeds through a vesicular phase, involving matrix vesicle-mediated nucleation of calcium phosphate, followed by a fibrillar phase, where mineral crystals grow within and along collagen fibers, contributing to the structural organization and mechanical properties of bone tissue [111].

Mineralization is a process restricted to terminally differentiated osteoblasts, either following cell differentiation or by exposing stable terminally differentiated cell lines (e.g., Saos-2) to differentiation media. Mineralization generally becomes evident after two to four weeks, depending on the cell type and experimental conditions [63]. The most widely used method for assessment and quantification of mineralization is Alizarin Red S (ARS) staining, which specifically binds to calcium deposits in the extracellular matrix [112]. After staining, the dye can be extracted and quantified spectrophotometrically. This provides a direct measure of calcium deposition [63]. Von Kossa staining is another classical method, primarily used to detect phosphate accumulation through the precipitation of silver phosphate, which appears as black deposits under light microscopy. While von Kossa staining is primarily qualitative, image analysis software can be employed to quantify the stained area, providing semi-quantitative results [113]. In addition to matrix staining methods, alkaline phosphatase (ALP) activity assays are commonly used to assess early osteogenic differentiation and cells’ capacity to promote mineralization. ALP plays a crucial role in mineralization by hydrolyzing inorganic pyrophosphate, an inhibitor of hydroxyapatite formation, thus increasing the availability of free phosphate ions necessary for mineral deposition. ALP activity is typically measured using colorimetric assays [114]. Further quantitative insights into mineralization can be obtained by directly measuring calcium content using colorimetric assays, such as the o-cresolphthalein complexone method, or phosphate concentration using molybdate-based colorimetric assays [115]. The reverse process of mineralization is bone matrix resorption, which is performed by osteoclasts. Functionally, osteoclast activity can be evaluated by in vitro bone resorption assays, in which differentiated osteoclasts are cultured on dentin or calcium phosphate-coated substrates. These systems allow visualization and quantification of resorption pits, offering a direct readout of osteoclast function [116,117].

### 5.4. Animal Models for Bone Research

Animal models play an essential role in bone research, particularly for studying complex diseases like osteoporosis, where interactions between multiple cell types, tissues, and systemic factors influence disease development. Unlike in vitro cell culture systems, animal models represent a fully functional, interconnected biological system, providing a more accurate approximation of physiological conditions. They enable researchers to investigate gene functions in the context of a living organism, including developmental, hormonal, and environmental influences that cannot be replicated in isolated cell cultures [118]. A key advantage of animal models is the ability to introduce targeted genetic modifications, something that is neither ethical nor possible in humans. While human studies rely on observing naturally occurring mutations, animal models allow precise manipulation of genes, including complete knock-outs, knock-ins of specific mutations or SNPs, and controlled overexpression [118]. This enables the study of gene function throughout all stages of life, including embryonic development, childhood growth, and adulthood—an important consideration, as some genes are only active during specific developmental periods, while others may play critical roles later in life. A classic example is the *Wnt1* gene, whose effects are highly dependent on developmental timing [119].

Mouse models remain the most established platform for investigating skeletal diseases due to their genetic similarity to humans and the availability of advanced genome editing technologies. CRISPR/Cas9 is used to generate knock-out and knock-in mouse models to study the effects of gene deletion or specific mutations on bone development, density, and strength. Humanized mouse models, which carry human genomic sequences, further enhance the relevance of these studies by allowing investigation of human gene regulation and variant effects in a living organism. However, despite their advantages, mouse models are costly and time-consuming to develop, and certain aspects of murine bone physiology differ from humans [120,121].

Although rat and mouse models are expensive to genetically manipulate, the zebrafish (Danio rerio) offers rapid and cost-effective genetic manipulation and has emerged as a valuable model organism for osteoporosis research. One of the unique advantages of zebrafish is their transparent embryos, which allow for easy visualization of skeletal development. Zebrafish are also capable of regenerating tissues and organs, including skeletal tissue. Using fluorescent microscopy, it is possible to study de novo bone formation after an induced fracture [122,123]. The expression of target genes and osteoblast-specific markers, such as Runx2, can be monitored during the recovery phase [124]. Zebrafish have gained popularity due to their genetic similarity to humans (71% of genes are conserved) and their transparency, which allows for non-invasive observation of skeletal development. Zebrafish models permit targeted gene knockdown or overexpression experiments, enabling the investigation of specific gene impacts on bone development and maintenance. The CRISPR/Cas9 system allows for relatively easy and efficient genome editing in zebrafish. First-generation “crispants” are often sufficient to phenotypically recapitulate knock-outs, which enables faster functional characterization of target genes [123]. Zebrafish models bridge the gap between GWAS findings and clinical applications by providing insights into the molecular mechanisms of skeletal diseases. Despite their advantages, zebrafish also have limitations, such as the absence of long bones and bone marrow, which restricts direct comparison with the human skeletal system. Nevertheless, by combining genetic, cellular, and pharmacological approaches, zebrafish offer a powerful platform for advancing our understanding and treatment of skeletal diseases [123].

WNT16, one of the most frequently implicated genes in BMD-related GWAS loci, plays a critical role in skeletal development through modulation of the Wnt signaling pathway [125,126]. To identify the molecular mechanism by which Wnt16 affects skeletal development and homeostasis, Qu et al. performed RNA sequencing analysis. Knock-out of the Wnt16 gene impacted Wnt signaling by inhibiting the mTOR and FoxO pathways, resulting in disruption of zebrafish bone development [127]. Watson et al., who showed that Wnt16 also affects muscle tissue, further demonstrated the pleiotropic effect of Wnt16. Wnt16 knock-out zebrafish exhibited impaired myogenesis and muscle growth. Overall, Wnt16 plays a critical role in maintaining musculoskeletal homeostasis [128]. The neural epidermal growth factor-like (EGFL)-like protein (NELL)-1 gene has frequently appeared in GWASs related to osteoporosis [129]. Studies have shown that NELL-1 stimulates the differentiation of mesenchymal stem cells into osteoblasts [130,131]. James et al. introduced point mutations into the Nell-1 gene in mouse models, resulting in transcript loss. Complete protein loss was lethal, whereas heterozygous mice exhibited normal skeletal development but developed age-related osteoporosis. Osteoblastogenesis was impaired, and excessive osteoclastogenesis was observed. Administration of recombinant NELL-1 increased osteogenic differentiation via activation of Wnt/β-catenin signaling and reduced bone resorption [132].

### 5.5. In Situ Tissue Gene Expression

Tissue isolation from patients is a crucial step in studying osteoporosis, as it allows for direct investigation of bone tissue characteristics and molecular changes associated with the disease. While data gained directly from human tissue is most relevant and exact, such tissues are difficult to obtain. Ethical considerations and institutional guidelines must be followed to ensure patient privacy and welfare. For tissue sampling, patients diagnosed with osteoporosis must be recruited. The selection criteria may include factors such as age, sex, bone mineral density, medical history, and medication use. Prior to tissue collection, patients must provide informed consent, fully understanding the purpose, risks, and potential benefits of the study. Tissue can be obtained through surgical procedures or minimally invasive biopsies. The choice of approach depends on the specific research objectives, accessibility of the target bone, and patient-related factors. Commonly sampled bones include the femur, vertebrae, iliac crest, or tibia. By isolating bone tissue from patients with osteoporosis, various analyses can be performed, including histological examination, gene expression profiling, proteomic studies, and stem cell isolation. Following tissue isolation, sample processing must be tailored to the intended downstream analyses. For histological studies, bone samples require decalcification to allow sectioning. This process, typically using EDTA or acid-based solutions, can be time-consuming and must be carefully optimized to preserve tissue morphology and molecular integrity [133]. Improper decalcification may result in poor staining quality or RNA degradation. Histological analyses can reveal critical information about bone microarchitecture, cellular composition, and pathological features. Common staining methods include Hematoxylin and Eosin (H&E) for general morphology, Masson’s Trichrome for collagen deposition, Safranin O for cartilage remnants, and tartrate-resistant acid phosphatase (TRAP) staining for osteoclast activity [134]. For molecular studies, high-quality RNA and DNA extraction from bone tissue remains technically challenging due to the dense and mineralized matrix. Mechanical disruption using cryogenic grinding or specialized homogenizers is often required. RNA isolation is particularly sensitive to degradation; thus, rapid sample processing and the use of RNA stabilization reagents are essential [135,136]. In cases where gene expression is analyzed, quantitative PCR or RNA sequencing can be performed to profile osteogenic and osteoclastic markers, such as RUNX2, OCN, RANKL, and OPG. Additionally, DNA methylation studies provide insights into epigenetic regulation of bone-related genes. Bisulfite conversion and next-generation sequencing platforms are commonly used to assess methylation patterns associated with osteoporosis progression [137]. Advanced imaging techniques, such as micro-computed tomography (microCT), are invaluable for the non-destructive assessment of bone microarchitecture. MicroCT provides high-resolution, three-dimensional data on bone density, trabecular thickness, and structural integrity, enabling precise quantification of osteoporosis-related changes [138,139]. When combined with histological and molecular analyses, microCT contributes to a comprehensive understanding of both structural and functional alterations in bone tissue.

Studying muscle [140] and adipose tissue alongside bone tissue can provide a more comprehensive understanding of the mechanisms of action of the studied gene [141]. In GWAS studies, DAAM2 was identified as an important gene for further functional characterization [19]. DAAM2 is involved in the regulation of canonical Wnt signaling. CRISPR/Cas knock-out of DAAM2 in mice led to reduced bone mass and increased porosity. DAAM2 is also expressed in human skeletal muscle, demonstrating its pleiotropic effect [19].

## 6. Future Perspectives

Recent progress in functional genomics—such as CRISPR-based perturbation screens, single-cell transcriptomics, and chromatin profiling [142,143]—has enabled more detailed characterization of genes and regulatory elements implicated by GWAS (Table 5). Yet, interpreting the wealth of diverse and often context-specific data remains a significant challenge. To address this, artificial intelligence (AI) is increasingly being employed to integrate genomic, epigenomic, and transcriptomic information. These tools can predict which variants are most likely to influence gene regulation, even in cell types not yet experimentally profiled, by learning patterns directly from sequence data [144].

Looking forward, the field will benefit from standardized approaches for evaluating gene causality and integrating emerging tools such as spatial transcriptomics, organ-on-chip systems, and AI-driven analytics. Together, these strategies are expected to accelerate the translation of genetic associations into biological understanding and clinical applications in complex traits like osteoporosis.

## Figures and Tables

**Figure 1 ijms-26-07237-f001:**
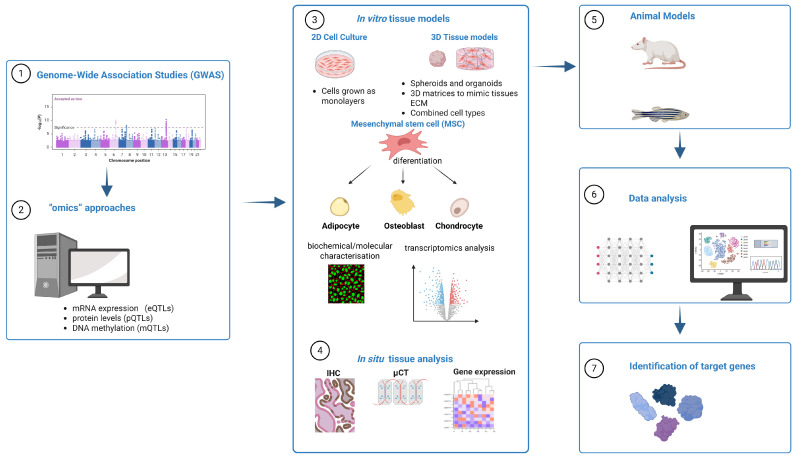
Workflow for genome-wide association studies (GWASs) folloved by functional characterization of identified genes. The process begins with the identification of single-nucleotide polymorphism (SNP) variants through GWAS (1). This is followed by in silico integrative “omics” approaches, which aid in the prioritization of functional loci based on datasets such as expression quantitative trait loci (eQTLs), protein quantitative trait loci (pQTLs), and DNA methylation quantitative trait loci (mQTLs) (2). Subsequently, candidate loci undergo in vitro functional characterization using both two-dimensional (2D) and three-dimensional (3D) tissue culture models (3). Complementary in situ analyses of patient tissue samples, including immunohistochemistry (IHC), provide additional validation (4). Functional evaluation of target genes is performed using animal models (5). Finally, integrated data analysis (6) across these experimental platforms enables the identification and confirmation of target genes involved in osteoporosis pathogenesis (7).

**Figure 2 ijms-26-07237-f002:**
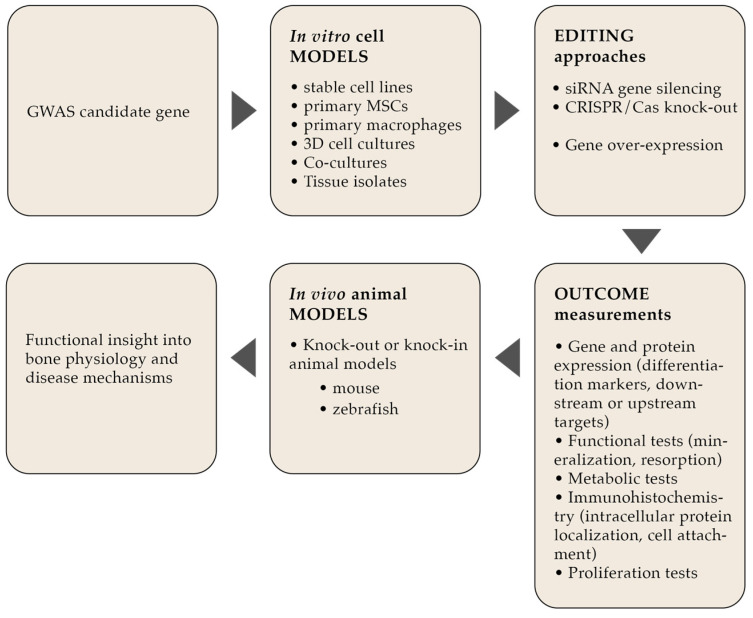
Systematic framework for functional characterization of GWAS hits.

**Table 1 ijms-26-07237-t001:** Commonly used cell models for in vitro bone research.

Cell Line	Species	Cell Type	Differentiation Potential	Mineralization	3D Culture Feasibility
HOS	Human	Osteoblast-like	Limited	Yes (under specific conditions)	Moderate
Saos-2	Human	Mature osteoblast-like	Limited	Yes	High
MG-63	Human	Pre-osteoblast	Limited	Low	Moderate
U2OS	Human	Osteosarcoma	Very limited	No	Low
MC3T3-E1	Mouse	Pre-osteoblast	High (osteoblast lineage)	Yes	High
RAW264.7	Mouse	Monocyte/macrophage	Differentiates into osteoclasts	Resorption	Moderate
THP-1	Human	monocyte	Differentiates into osteoclasts	Resorption	Moderate
MSCs (primary)	Human	Mesenchymal stem cell	Osteogenic, chondrogenic, adipogenic	Yes (after induction)	High
hFOB 1.19	Human	Immortalized osteoblast	High at permissive temperature	Yes	Moderate
CAL-72	Human	Osteosarcoma	Limited	No	Low
TE-85	Human	Osteosarcoma	Limited	No	Low

**Table 2 ijms-26-07237-t002:** Comparison of commonly used experimental cell and animal models for functional characterization of GWAS hits in bone research.

Model	Advantages	Limitations	Recommended Applications
Primary MSCs (bone-derived)	High physiological relevance; multilineage differentiation potential	Limited proliferation; donor variability; difficult to genetically manipulate	Functional validation of osteogenic genes gene expression profiling; differentiation studies
Primary monocytes (blood- or bone-derived)	Easily accessible; physiologically relevant; can generate mature osteoclasts	Fragile; difficult to genetically manipulate; batch variability	Osteoclastogenesis assays; gene expression; TRAP activity studies
Immortalized osteoblast-like and monocyte cell lines	Easy handling; unlimited proliferation; transfection and gene editing	Altered phenotype; reduced mineralization potential lower physiological relevance	Initial mechanistic screening; siRNA/shRNA or CRISPR studies; gene overexpression/knockdown
3D spheroid	Mimics tissue-like environment; enhances osteogenesis; allows co-culture setups	Technically demanding; lower throughput; limited standardization	Bone remodeling studies; osteoblast–osteoclast interaction; scaffold testing
Mouse models (knock-out, knock-in, transgenic)	Whole-organism context; skeletal phenotype assessment; strong genetic tools	High cost; time-consuming	In vivo validation of gene function; developmental and systemic effect studies
Zebrafish (knock-out, knock-in, transgenic)	Transparent embryos; rapid bone development; easy genetic manipulation; regeneration studies	Lack of long bones and bone marrow; limited translational equivalence	Fast in vivo gene function screening; developmental studies; skeletal regeneration assays

**Table 3 ijms-26-07237-t003:** Comparison of commonly used gene manipulation approaches for functional characterization of GWAS hits in bone research.

**Method**	**Advantages**	**Limitations**	**Recommended Applications**
CRISPR/Cas9 Knock-out	Permanent gene disruption; high specificity; enables loss-of-function studies	Off-target effects; requires clonal selection; may induce compensatory pathways	Functional validation of essential genes; early developmental pathway analysis
CRISPR interference (CRISPRi)	Reversible gene silencing; targets non-coding regions; no DNA cleavage	Requires stable dCas9 expression; incomplete silencing possible	Regulation of enhancers/promoters; dose-dependent gene suppression
CRISPR activation (CRISPRa)	Gene upregulation from endogenous locus; no need for cDNA overexpression	Efficiency depends on chromatin context; requires guide RNA design and dCas9 fusion systems	Functional gain-of-function studies; promoter/enhancer mapping
RNA interference (siRNA)	Fast, transient gene knockdown; easy to apply in most cell lines	Off-target effects; transient; may not fully deplete target mRNA	Initial screening; pathway studies; short-term gene function testing
shRNA (short hairpin RNA)	Stable knockdown via integration; allows long-term silencing	Time-consuming cloning; potential for off-target effects; variable expression	Long-term gene silencing in immortalized or primary cells
Plasmid overexpression	Easy to design; widely used; applicable to many cell lines	Non-physiological expression levels; transient in most systems	Gain-of-function studies; rescue experiments

**Table 4 ijms-26-07237-t004:** Key molecular markers of osteoblast differentiation, stages of expression, and functional roles in bone formation.

Marker	Stage of Expression	Function
SOX9	Early	Transcription factor marking mesenchymal precursors
RUNX2	Early to intermediate	Master regulator of osteoblast differentiation
ALP	Intermediate	Enzyme involved in the onset of matrix mineralization
COL1A1	Early to late	Major structural protein of bone extracellular matrix
OSX (SP7)	Intermediate to late	Essential transcription factor for osteoblast maturation
OCN (BGLAP)	Late	Marker of mature osteoblasts; involved in bone mineralization
OPN (SPP1)	Late	Mediates cell adhesion and matrix remodeling
BSP	Late	Binds calcium; important for initial stages of mineral deposition
DLX5	Early to intermediate	Promotes osteogenesis via signaling pathways such as Notch
ATF4	Intermediate to late	Regulates osteoblast function and inhibits osteoclast differentiation

**Table 5 ijms-26-07237-t005:** Recently functionally evaluated genetic variants associated with OP. Summary of genes that were functionally characterized from 2020 to the present.

Gene	Study Approach	Function	Reference
*ANAPC1*	-The expression of the ANAPC1 gene was examined in the human bone and muscle tissue samples from osteoporotic, osteoarthritic, and healthy individuals by quantitative PCR (q-PCR) -Osteogenic and adipogenic differentiation of MSCs -Silencing of ANAPC1 in HOS cells	ANAPC1 plays a role in bone physiology and osteoporosis development, with decreased expression in osteoporotic patients and altered expression during osteogenic differentiation of human mesenchymal stem cells.	[55]
*CCDC170*	-Cloning of the different SNP alleles into a luciferase reporter vector, transfecting cells with the vectors along with miRNA mimics/inhibitors, and performing luciferase reporter assays -RNA isolation, cDNA synthesis, and qRT-PCR to measure gene expression levels -ELISA assays to measure protein levels of osteogenesis and osteoclastogenesis markers -In vivo mouse experiments with CCDC170 knockdown	The CCDC170 gene, through its interaction with microRNAs and specific genetic polymorphisms, plays a significant role in bone health and the risk of osteoporosis.	[56]
*LRP5*	CRISPR/Cas9 gene editing, using a gRNA with high predicted out-of-frame efficiency	LRP5 acts as a co-receptor in the Wnt signaling pathway, binding Wnt ligands and interacting with Frizzled. Loss of LRP5 function leads to impaired Wnt signaling and reduced osteoblast differentiation.	[145]
*USF3*	Overexpression and knockdown in U-2OS cells, luciferase reporter assay, biotin pull-down	Transcriptional regulator of osteogenesis and osteoclastogenesis.	[54]
*EPDR1*	-CRISPR-Cas9 genome editing in osteoblast cells to delete the region containing the BMD-associated variants -Measurement of EPDR1 gene and protein expression in the edited cells using RT-qPCR and Western blotting -Assessment of alkaline phosphatase activity, a marker of osteoblast differentiation, in the edited cells	EPDR1 plays a key role in osteoblast differentiation and bone mineral density determination.	[146]
*miR-199a-5p*	Overexpression of miR-199a-5p in human mesenchymal stem/progenitor cells	miR-199a-5p regulates the terminal fate specification of MSCs into osteoblasts or chondrocytes, with overexpression favoring chondrogenic differentiation.	[61]
*PPP6R3*	Deletion of Ppp6r3 gene in mice, TWAS/colocalization approach using GTEx	PPP6R3 is a regulatory subunit of protein phosphatase 6. Ppp6r deletion in mice decreased BMD.	[147]
*SPTBN1*	Single-cell RNA sequencing	SPTBN1 is a cytoskeleton protein that contributes to organ development by establishing and maintaining cell structure and regulating various cellular functions. It is also involved in bone structure development and fracture healing.	[148]
*FAM210A*	Fam210a knock-out mice, to study the effects of Fam210a on bone and muscle biology -X-Gal staining to detect Fam210a expression in mouse tissues -Phenotypic analyses in the mouse models, including measurements of bone mineral density, bone biomechanical properties, muscle function, and gene expression	The function of the FAM210A protein is to regulate both bone and muscle structure and function.	[149]

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
