# Peer review of "Recent Advances in Experimental Functional Characterization of GWAS Candidate Genes in Osteoporosis"

_ijms, 2025, doi:10.3390/ijms26157237_

Round 1

Reviewer 1 Report

Comments and Suggestions for Authors

Manuscript ID: ijms-3688706

Title: Recent advances in experimental functional characterization of GWAS candidate genes in osteoporosis

This review focuses on current methodologies and key examples of successful functional studies aimed at evaluating gene function in osteoporosis research. but there are some problems, the manuscript also should be revised due to many major and minor issues. The detailed explanation is as follows:

Major comment

  1. The review is missing a strong central message and unique contribution.
  • The introduction explains why functional studies after GWAS are important, especially because many SNPs are in non-coding regions. This is true, but it's also already well known in the field.

  • What is not clear is:

 Why this review is needed now,

 What makes it different from other reviews, and

 What new ideas or insights it brings.

A stronger central narrative and clearer positioning would significantly enhance the impact of the manuscript.

  1. Lack of Practical Guidance for Model Selection

.-  Although various experimental approaches (in silico, in vitro, in vivo, in situ) are described, the manuscript does not provide guidance on how to choose the most appropriate model depending on the gene of interest or experimental goal.
A comparative summary or decision-making framework would increase the review's utility for researchers.

Minor comment

  1. Inconsistent use of abbreviations
    Terms such as MSC, hMSC, and MSCs are used interchangeably. Please define each abbreviation upon first mention and use them consistently throughout the manuscript.
  2. Typographical and spelling errors
    There are several typographical errors, including:\

“age ralated” → “age-related”

“physycal inactivity” → “physical inactivity”

“Unfortunately, most 0of the SNPs” → 0 (x)

  1. Weak “Future Perspectives” section
    The final section is too brief and lacks specificity. Consider expanding it with more concrete insights into emerging tools (e.g., CRISPRi, single-cell analysis) and directions for future research.

Author Response

Major comment:

  1. The review is missing a strong central message and unique contribution. The introduction explains why functional studies after GWAS are important, especially because many SNPs are in non-coding regions. This is true, but it's also already well known in the field. What is not clear is: Why this review is needed now, what makes it different from other reviews, and what new ideas or insights it brings. A stronger central narrative and clearer positioning would significantly enhance the impact of the manuscript.

Response:
We thank the reviewer for pointing out the lack of adequate explanation of the goal and novelty of this review. Our main goal was to create an overview and methodological framework for someone starting in the complex field of functional characterization of novel genes associated with bone biology and osteoporosis -  a document we wish we had when we first started. This review aims to bridge the methodological gap between genetic variations discovery and biological validation in osteoporosis research by providing a clear guidance and goal-oriented framework to help researchers select appropriate tools for functional gene validation tailored to the context of bone biology. We provide methodological guidance and experience accumulated through years of experimental work and optimization of the methods to adequately asses the bone-related effects of the observed gene/protein.

To emphasise this, we have rewritten the last paragraph of the introduction, more clearly explaining our goal:

“The goal of this review is to examine current methodologies and key examples of the successful functional characterization of GWAS-identified target candidate genes in osteopo-rosis research. With this work, we aim to bridge the methodological gap between genetic variations discovery and biological validation in osteoporosis research by providing a clear guidance and goal-oriented framework to help researchers select appropriate tools for functional gene validation tailored to the context of bone biology. By leveraging the methodological overview and practical examples presented in this review, scientists will be better equipped to choose and combine experimental strategies that accelerate the dis-covery of how specific genes and their protein products influence bone cell physiology, contribute to skeletal tissue homeostasis, and ultimately play a role in the development of complex disorders such as osteoporosis (9,10). With this we hope to facilitate and en-courage the urgently needed experimental studies on confirmation and functional char-acterization of candidate genes that move beyond statistical associations and prediction models. Importantly, this review specifically focuses specifically on the evaluation of gene function itself—independent of variant-specific regulatory effects—and does not address functional studies of non-coding regulatory variants, transcript isoforms, or vari-ant-driven changes in gene expression. These topics are comprehensively covered in other reviews (11–13).”

  1. Lack of Practical Guidance for Model Selection .- Although various experimental approaches (in silico, in vitro, in vivo, in situ) are described, the manuscript does not provide guidance on how to choose the most appropriate model depending on the gene of interest or experimental goal. A comparative summary or decision-making framework would increase the review's utility for researchers.

Response:

We thank the reviewer for this valuable and constructive comment. In response, we have added Figure 2 entitled "Systematic framework for functional characterization of GWAS hits", which outlines a stepwise decision-making process for functional evaluation of target genes. Furthermore, we have included three additional comparative tables to enhance the manuscript’s practical value:

  • Table 2: Comparison of commonly used experimental cell and animal models for functional characterization of GWAS hits in bone research
  • Table 3: Comparison of commonly used gene manipulation approaches for functional characterization of GWAS hits in bone research
  • Table 4: Key molecular markers of osteoblast differentiation, stages of expression, and functional roles in bone formation

These additions aim to provide clearer guidance and a more comprehensive overview for researchers selecting appropriate tools and models for functional characterization.

Minor comment:

Minor comment 1. Inconsistent use of abbreviations Terms such as MSC, hMSC, and MSCs are used interchangeably. Please define each abbreviation upon first mention and use them consistently throughout the manuscript.

Response: We thank the reviewer for pointing out the inconsistent use of abbreviations. We have carefully revised the manuscript and now use the term MSC (mesenchymal stromal cell) consistently throughout. The abbreviation is defined upon its first mention, and 21 instances of inconsistent usage have been corrected.

Minor comment 2: Typographical and spelling errors There are several typographical errors, including:\“age ralated” → “age-related”, “physycal inactivity” → “physical inactivity”, “Unfortunately, most 0of the SNPs” → 0 (x)

Response: We thank the reviewer for noting these typographical errors. We have thoroughly proofread the manuscript and corrected all identified spelling and typographical mistakes, including those mentioned by the reviewer. The revised text has been checked throughout to ensure clarity and accuracy.

Minor comment 3: Weak “Future Perspectives” section

The final section is too brief and lacks specificity. Consider expanding it with more

concrete insights into emerging tools (e.g., CRISPRi, single-cell analysis) and

directions for future research.

Response:

We appreciate the reviewer’s suggestion to expand the “Future Perspectives” section. In response, we have substantially revised this part of the manuscript to provide a more detailed outlook on emerging methodologies and research directions (see Chapter 6). We hope this expanded section better conveys the exciting directions in which the field is moving and underscores the transformative potential of these advanced tools.

“Recent progress in functional genomics—such as CRISPR-based perturbation screens, single-cell transcriptomics, and chromatin profiling (149,150)—has enabled more detailed characterization of genes and regulatory elements implicated by GWAS. Yet, interpreting the wealth of diverse and often context-specific data remains a significant challenge. To address this, artificial intelli-gence (AI) is increasingly being employed to integrate genomic, epigenomic, and transcriptomic information. These tools can predict which variants are most likely to influence gene regulation, even in cell types not yet experimentally profiled, by learning patterns directly from sequence data (151). Looking forward, the field will benefit from standardized approaches for evaluating gene cau-sality and integrating emerging tools such as spatial transcriptomics, organ-on-chip systems, and AI-driven analytics. Together, these strategies are expected to accelerate the translation of genetic associations into biological understanding and clinical applications in complex traits like osteoporosis.”

Reviewer 2 Report

Comments and Suggestions for Authors

Overall an interesting update on the developments in the field of osteoporosis omics. The manuscript is well developed though I would have expected some pathway approaches in developing the narrative for this paper to give a clear focus and future developments. Main techniques and tools have been described adequately with their potential and limitations. The future perspectives section should be further developed and expanded, as it is currently very generic and lacks specificity to potential targets and possibilities for BMD and osteoporosis. It would also be interesting to document,  if there were gender differences in osteoporosis  GWASs as individual/gender physiology may have variable expression and functionalities. 

Section 5 would benefit from inclusion of some relevant references.

Some formatting is needed on page 8 as in the middle paragraph, a different font is used/edited.

Author Response

Overall an interesting update on the developments in the field of osteoporosis omics. The manuscript is well developed though I would have expected some pathway approaches in developing the narrative for this paper to give a clear focus and future developments:

We thank the reviewer for the positive evaluation and the thoughtful suggestion. In response, we have expanded the manuscript to better reflect functional pathway by including Figure 2, titled "Systematic framework for functional characterization of GWAS hits".

Main techniques and tools have been described adequately with their potential and limitations. The future perspectives section should be further developed and expanded, as it is currently very generic and lacks specificity to potential targets and possibilities for BMD and osteoporosis. It would also be interesting to document,  if there were gender differences in osteoporosis  GWASs as individual/gender physiology may have variable expression and functionalities:

We thank the reviewer for this comment and agree that sex-specific differences in osteoporosis genetics are an important aspect that warrants inclusion. We have now expanded the “Future Perspectives” section to address this point (see Chapter 6).

“Recent progress in functional genomics—such as CRISPR-based perturbation screens, single-cell transcriptomics, and chromatin profiling (149,150)—has enabled more detailed characterization of genes and regulatory elements implicated by GWAS. Yet, interpreting the wealth of diverse and often context-specific data remains a significant challenge. To address this, artificial intelligence (AI) is increasingly being employed to integrate genomic, epigenomic, and transcriptomic information. These tools can predict which variants are most likely to influence gene regulation, even in cell types not yet experimentally profiled, by learning patterns directly from sequence data (151). Looking forward, the field will benefit from standardized approaches for evaluating gene causality and integrating emerging tools such as spatial transcriptomics, organ-on-chip systems, and AI-driven analytics. Together, these strategies are expected to accelerate the translation of genetic associations into biological understanding and clinical applications in complex traits like osteoporosis.”

Section 5 would benefit from inclusion of some relevant references.

We appreciate the reviewer’s suggestion. In response, Section 5 was substantially expanded to provide a more comprehensive overview of strategies used in the functional characterization of GWAS genes, with a clear focus on their application in bone biology. The revised section now includes relevant references supporting the conceptual framework and examples mentioned throughout the chapter. Additionally, we provide a visual summary in Figure 2 and further detail these approaches in newly added Tables 2–4, to guide researchers in selecting appropriate experimental models, gene perturbation techniques, and readouts for assessing bone-specific outcomes.

Some formatting is needed on page 8 as in the middle paragraph, a different font is used/edite

We thank the reviewer for noticing this formatting issue. The font inconsistency on page 8 has been corrected, and the entire manuscript has been reviewed for uniform formatting.